# Consequences and Mitigation Strategies of Abiotic Stresses in Wheat (*Triticum aestivum* L.) under the Changing Climate

Akbar Hossain [1,*], Milan Skalicky [2,*], Marian Brestic [2,3], Sagar Maitra [4], M. Ashraful Alam [5], M. Abu Syed [6], Jamil Hossain [7], Sukamal Sarkar [8], Saikat Saha [9], Preetha Bhadra [10], Tanmoy Shankar [10], Rajan Bhatt [11], Apurbo Kumar Chaki [12,13], Ayman EL Sabagh [14,15] and Tofazzal Islam [16]

1   Bangladesh Wheat and Maize Research Institute, Dinajpur 5200, Bangladesh
2   Department of Botany and Plant Physiology, Faculty of Agrobiology, Food, and Natural Resources, Czech University of Life Sciences Prague, Kamycka 129, 165 00 Prague, Czech Republic; marian.brestic@uniag.sk
3   Department of Plant Physiology, Slovak University of Agriculture, Nitra, Tr. A. Hlinku 2, 949 01 Nitra, Slovakia
4   Department of Agronomy, Centurion University of Technology and Management, Paralakhemundi 761211, Odisha, India; sagar.maitra@cutm.ac.in
5   Plant Breeding Division, Spices Research Centre, Bangladesh Agricultural Research Institute, Shibganj, Bogura 5880, Bangladesh; a.alam_83@yahoo.com
6   Plant Breeding Division, Bangladesh Rice Research Institute, Gazipur 1701, Bangladesh; msyedso@yahoo.com
7   Agronomy Division, Regional Agricultural Research Station (RARS), BARI, Ishordi, Pabna 6620, Bangladesh; jamilbari11@gmail.com
8   Department of Agronomy, Bidhan Chandra Krishi Viswavidyalaya (BCKV), Nadia 741101, West Bengal, India; sukamalsarkarc@yahoo.com
9   Nadia Krishi Vigyan Kendra, Bidhan Chandra Krishi Viswavidyalaya (BCKV), Gayeshpur, Nadia 741234, West Bengal, India; saikatsaha2012@gmail.com
10  Department of Biotechnology, Centurion University of Technology and Management, Paralakhemundi 761211, Odisha, India; preetha.bhadra@cutm.ac.in (P.B.); tanmoy@cutm.ac.in (T.S.)
11  Regional Research Station, Kapurthala, Punjab Agricultural University, Ludhiana 141001, Punjab, India; rajansoils@pau.edu
12  School of Agriculture and Food Sciences, The University of Queensland, Brisbane, QLD 4072, Australia; a.chaki@uq.net.au
13  Bangladesh Agricultural Research Institute, Gazipur 1701, Bangladesh
14  Department of Agronomy, Faculty of Agriculture, Kafrelsheikh University, Kafrelsheikh 33512, Egypt; aymanelsabagh@gmail.com
15  Department of Field Crops, Faculty of Agriculture, Siirt University, 56100 Siirt Merkez/Siirt, Turkey
16  Institute of Biotechnology and Genetic Engineering (IBGE), Bangabandhu Sheikh Mujibur Rahman Agricultural University, Gazipur 1706, Bangladesh; tofazzalislam@yahoo.com
*   Correspondence: akbarhossainwrc@gmail.com (A.H.); skalicky@af.czu.cz (M.S.)

**Abstract:** Wheat is one of the world's most commonly consumed cereal grains. During abiotic stresses, the physiological and biochemical alterations in the cells reduce growth and development of plants that ultimately decrease the yield of wheat. Therefore, novel approaches are needed for sustainable wheat production under the changing climate to ensure food and nutritional security of the ever-increasing population of the world. There are two ways to alleviate the adverse effects of abiotic stresses in sustainable wheat production. These are (i) development of abiotic stress tolerant wheat cultivars by molecular breeding, speed breeding, genetic engineering, and/or gene editing approaches such as clustered regularly interspaced short palindromic repeats (CRISPR)-Cas toolkit, and (ii) application of improved agronomic, nano-based agricultural technology, and other climate-smart agricultural technologies. The development of stress-tolerant wheat cultivars by mobilizing global biodiversity and using molecular breeding, speed breeding, genetic engineering, and/or gene editing approaches such as CRISPR-Cas toolkit is considered the most promising ways for sustainable wheat production in the changing climate in major wheat-growing regions of the world. This comprehensive review updates the adverse effects of major abiotic stresses and discusses the potentials of some novel approaches such as molecular breeding, biotechnology and genetic-engineering, speed breeding, nanotechnology, and improved agronomic practices for sustainable wheat production in the changing climate.

**Keywords:** abiotic stresses; mitigation strategies; wheat; global warming; CRISPR-Cas technology

## 1. Introduction

The extreme events of abiotic stresses such as drought, high temperature, salinity, etc. reduce crop production worldwide. Among them, the rise of atmospheric temperature due to anthropogenic activities significantly influences the function of agro-ecosystems. The Intergovernmental Panel on Climate Change (IPCC) [1] estimated that if anthropogenic activities continue to contribute to global warming in the current trend, there will be a chance of an increase in the earth's temperature by 6.4 °C. Consequently, the melting of glaciers enhances sea level rise up to 59 cm by the end of the 21st century. Climate change enhances the possibility of increasing different natural calamities such as floods, drought, storms, cyclones, and changes in the pattern of precipitations. As agriculture is highly climate dependent and sensitive to agro-climatic situations, the variations in temperature, humidity, and rainfall adversely impact the productivity of crop plants [2–4].

Among the cereals, wheat is the vital staple that provides around 20% of the calories and >25% of the protein to humans [5]. It occupies 30% of world cereal production (734 million tons) from 214 million hectares [6]. It provides quite satisfactory level of dietary fiber [7] and is considered a nutritious food grain [8,9]. Future predictions suggested that there will be an increased demand for wheat by about 60% in 2050 to feed an estimated 9.7 billion population in the world [10]. However, during the last decade, wheat productivity enhancement was not satisfactory to meet the future demand as it was increased by only 1.1% [11]. Recently, several international organizations such as IPCC, International Maize and Wheat Improvement Centre (CIMMYT)-International Centre for Agricultural Research in Dry Areas (ICARDA), and The Organization for Economic Co-operation and Development (OECD)-Food and Agriculture Organization (FAO) have forecasted that the extreme events of abiotic stresses such as drought, high temperature, salinity, etc. will reduce wheat yield by 20–30% especially in the developing countries. Additionally, global climate change has caused various biotic (such as wheat blast disease [12,13]) and hostile environmental events such as drought, thermo-stress, erratic rain, hailstorm, and salinity which instigated yield loss. Furthermore, a meta-analysis indicated that yield loss may occur with the rise of every 2 °C of temperature in sub-tropical and temperate regions [14]. In another study, Asseng et al. [15] predicted that there will be a 6% decline in wheat production with every 1 °C of temperature increase, which is estimated equivalent to about 42 million tons of wheat.

Among different abiotic stresses, the severity of the drought is remarkably impacted on the productivity of wheat [16]. Drought stress causes damage during all the stages of crop growth. If drought is occurred at the early growth stage of wheat, poor seedling stand establishment and less number of tillers per unit area are developed. However, drought incidence at the mid-stage of growth causes reduced dry matter production, effective tillers, and grains per plant [17]. The effect of drought at the terminal growth stage is also fatal in wheat as it reduces assimilate production, fertility, and grain weight [18]. Wheat requires an optimum temperature of 14 to 15 °C at the ripening stage and the temperature above 25 °C reduces grain weight [19], but the critical temperature for grain filling is 35.4 °C [20]. Under the changing climate, fluctuation of diurnal temperature causes yield loss [21,22]. The temperature rises above the threshold level adversely affect grain formation and filling [23], grain yield per spike [24], and yield loss [25,26].

Salinity is another major abiotic stress that adversely influences wheat yield and quality [27]. In wheat, salinity stress affects yield attributes and reduces grain weight, spike length, spikelet number, and grain yield [26]. The occurrence of natural calamity, untimely rain, flooding, and storm are becoming common nowadays, and these erratic weather threats to crop yield including wheat [28]. India is the world's second-highest wheat producing country in the world. The early cessation of rains is common in the eastern

and north-eastern part of India at the time of wheat maturity (March), causing pre-harvest sprouting [26], which ultimately decreases the quality due to early α-amylase enzyme activity with less Hagberg falling number [29]. Further, excess and untimely rain cause crop lodging and waterlogging, which negatively influence all crops including wheat in terms of productivity and grain quality [30]. Under high fertility conditions, wheat tends to lodge and natural calamities like hailstorms and cyclones damage the crop severely [31]. Various abiotic stresses are also threatening to wheat production in other major wheat-growing countries, i.e., China, the USA, Australia, Russia, Pakistan, and others.

Although a huge number of earlier studies highlighted the impacts of environmental stresses on growth, physiological changes, and yield of wheat, an inclusive overview is restricted for updating the consequences of abiotic stress and their possible mitigation strategies for wheat to meet the food security of increasing population [32]. This review updates current knowledge of the impacts of abiotic stresses on wheat and focuses on novel mitigation strategies for climate-smart wheat production for ensuring food and nutritional security of the ever-increasing global population.

## 2. Adaptive Mechanisms during Abiotic Stress in Wheat

When plants are unprotected from various environmental stresses such as drought, salinity, and heat stress, a large number of reactive oxygen species (ROS) are biosynthesized in the chloroplast, mitochondria, and/or peroxisomes of plant cells [33]. The ROS are highly reactive and thus higher production of ROS is capable of impairing normal plant metabolism. The high concentration of ROS can be injurious to the plant cells through enhancing lipid peroxidation, protein degradation, and DNA denaturation that can lead to oxidative damage, limit enzymatic activity, cellular arrangements, and even cell death [34,35]. Another stress metabolite, methylglyoxal (MG), is generated in plant chloroplasts, mitochondria, and cytosol under abiotic stresses, which is also toxic to plant cells. It creates oxidative stress and disrupts cellular function depending on the elevated amounts of MG and the severity of the stresses [36].

Among the abiotic stresses, drought stress creates inequality between antioxidant defenses and drought-induced ROS production. Tambussi et al. [37] observed that the wheat photosynthetic system is gradually damaged with the increasing period of water stress due to the generation of ROS in the chloroplast. The production of Malondialdehyde (MDA) has been suggested as the signal of ROS generation causing oxidative damage [38]. The elevated concentration of $H_2O_2$ and MDA in the cells of wheat plants break down the adaptive mechanism under drought-induced oxidative stress [39]. Similar to drought stress, salinity stress is also a major environmental constraint that inhibits plant growth and physiological activity through excessive production of ROS. A higher concentration of MDA accumulation in the cells of wheat cultivars under salinity stress causes oxidative damage in parenchymal cells of wheat plants [40]. Masood et al. [41] observed that exogenous application of NaCl caused more oxidative damage in wheat seedlings due to the high concentration of MDA in the cells of wheat. Ashraf et al. [42] also demonstrated that the elevated amount of MDA content in cells adversely affects the growth of wheat cultivars under severe salinity stress. Similarly, heat stress induces the generation and aggregation of ROS in plant cells. Savicka and Škute [43] observed an increase in $O_2^-$ generation in the wheat seedling under heat stress that influences a higher amount of MDA production, which lead to seedlings death. Besides, Hasanuzzaman et al. [44] reported that the chlorophyll content in the wheat seedlings was varied due to lipid peroxidation and a higher level of $H_2O_2$ content in the cells under high-temperature stress.

### 2.1. Biochemical Adaptation under Abiotic Oxidative Stress

The plant produces various types of compatible organic solutes in response to abiotic stress [45]. Under stressful conditions, plants are acclimatized and modulate the usual metabolic pathways. During salinity and drought stresses, plant cells maintain water status in their tissues through osmotic adjustment. The osmotic adjustment involves the

biosynthesis of osmotically dynamic molecules such as proline, glycine betaine, sugar alcohols, soluble sugars, organic acids, chloride ions, calcium, and potassium. Among them, proline is the most common osmolyte and signaling compatible organic solute, which is biosynthesized normally in the cytosol and plastid but its degeneration occurs in mitochondria [46]. It provides many functions in plant cells including the stability of membrane and protein, buffering the cellular redox potential, quenching free radicals, and mitigates the adverse effect of oxidative stress in response to various abiotic stresses [47]. The proline is mainly produced in the plant cell from two pathways viz. the glutamate and ornithine pathways [48]. A major part of the proline synthesis occurs through the glutamate pathway. Two leading enzymes such as pyrroline 5-carboxylate synthetase (P5CS) encoded by two genes and pyrroline 5-carboxylate reductase (P5CR) encoded by one gene are involved to synthesize proline via the glutamate pathway [49]. The accumulation of proline is the first response to reduce cell injury when plants are exposed to water deficit stress. It does not impair enzymatic activity even at a higher concentration. The enhanced accumulation and mobilization of proline under drought stress showed an increased level of tolerance [50]. Similarly, Hong-Bo et al. [51] described that proline acted as an anti-drought defense compound. A positive association between drought-tolerance and proline content exists in wheat genotypes as observed by Mwadzingeni et al. [52]. Treatment of seeds with proline enhances the growth and yield of wheat [53,54]. Recently, Farooq et al. [55] illustrated that under drought stress, the total phenolics, chlorophyll, and proline contents were enhanced when proline was applied exogenously. They also observed that exogenous application of proline also increased the number of grain and grain weight. Proline application also enhances the growth and development of wheat plants under salinity-induced oxidative stress [56–58]. These results indicate that proline-induced adaptation to salt tolerance is associated with the enhancement of root and shoot length, fresh and dry weight of seedling, photosynthetic pigment, as well as $K^+$ content and $K^+/Na^+$ ratio [56–58].

Glycine betaine (GB) is an electrically neutral and dipolar compatible organic solute, which is primarily biosynthesized in the chloroplast. It plays an important role in the adjustment of chloroplast and safeguards the thylakoid membranes. The GB controls the photosynthetic competence and integrity of the plasma membrane [59]. The GB reduces salt-induced oxidative damage by minimizing $Na^+$ accumulation and maintain $K^+$ concentration in plant cells [60]. Under drought stress, GB also regulates the intracellular osmotic potential, cytoplasmic pH, structure of cell membrane and also protects the activities of the antioxidant enzymes [35]. Exogenous application of GB protects different organelles such as chloroplasts and mitochondria from stress-induced oxidative injury [35]. In higher plants, GB is synthesized from two substances like choline and glycine encoded by different enzymes. Although plants naturally accumulate low levels of GB, significant enhancement of the GB in plant cells helps plants to survive under the abiotic stresses [61]. The exogenous application of GB decreases proline synthesis in wheat plant cells that ultimately helps to maintain water balance and photosynthetic pigments under salinity stress [62]. In another study, Raza et al. [63] revealed that stomatal conductance and net $CO_2$ assimilation are enhanced due to exogenous application of GB on wheat under salt-stressed conditions. GB helps to improve photosynthetic capacity and osmotic adjustment in salt-affected plants by reducing salinity-induced oxidative damages in wheat [64]. Salama et al. [65] stated that foliar application of choline condensed the endogenous proline, MDA, and glutathione contents, while enhancement of GB, $K^+$, and $Ca^{2+}$ concentrations in shoots and roots of wheat plants maintained the membrane integrity. The GB application also promotes higher levels of glutathione and decreases MDA and proline contents in wheat seedlings that ultimately reduces salt-induced oxidative damage and maintains ion homeostasis and membrane stability under salt-stressed conditions [66]. Foliar application of GB enhances the synthesis of choline, proline, and sucrose in wheat cultivars under drought stress conditions [67].

Polyamines (PA) are small positively charged organic molecules. They provide membrane stability, regularize ionic and osmotic balance, and function as antioxidants through binding with DNA, RNA, protein, and phospholipids [68]. Polyamines are found in higher plants in the form of spermine (Spm), spermidine (Spd), and putrescine (Put). These molecules actively respond to different abiotic stresses such as salt, drought, and temperatures to enhance the photosynthetic capability, and maintain water balance through improving osmotic adjustment [69]. Liu et al. [70] reported that polyamines restrict water loss in plants through strongly regulating guard cell opening and closing. In addition, the size of the guard cells and the potassium gated channel are being determined by the polyamines. El-Shintinawy [71] demonstrated that the accumulation of Spm and Spd excessively increased whereas the amount of Put content was reduced in wheat cultivars under salinity stress. Liu et al. [72] observed significantly higher amounts of free Spm and Spd levels in the leaves of drought-tolerant wheat cultivars, whereas increased free putrescine levels in drought-sensitive wheat cultivars in response to PEG 6000 treatment. In another study, Liu et al. [73] investigated the effects of exogenous polyamines application on wheat genotypes under drought stress. They found enhanced free Spm and Spd but decreased Put concentration in the grains of wheat under drought stress. Similarly, Yang et al. [74] reported that the enhancement of Spd and Spm reduced drought-induced oxidative stress. A significant change in the stability of the plasma membrane is achieved by the effects of exogenous application of PA in wheat plants [75]. Sakr and El-Metwally [76] reported that exogenous spermine application reduces the negative effects of salt stress through enhancing the protein and PA contents in plant cells. Similarly, Asthir et al. [77] observed the effect of foliar use of Put on the seedlings of wheat under high-temperature stress (45 °C) and revealed that the Put application resulted in higher peroxidase (POX) and superoxide dismutases (SOD) events and augmented ascorbate and tocopherol contents in the grains of wheat. Furthermore, the enhancement of POX and SOD actions was determined after the increase of the levels of polyamine oxidase (PO), diamine oxidase (DAO), and catalase GR glutathione reductase (CAT). Recently, Jing et al. [78] experimented to know the impact of PA when applied exogenously on diverse heat-resistant and susceptible wheat varieties under heat stress. The exogenous application of Spd and Spm improved the survival ability of susceptible wheat varieties during grain filling stage of wheat [78].

Carbohydrate, particularly sucrose, is a photosynthetic product, which plays a significant role in growth and development of plants. During stress conditions, several carbohydrates act as osmoprotectant for osmotic adjustment, scavenge excessively produced ROS, defense integrity of cell membrane and DNA, and also stabilize essential enzymatic activity [79–81]. For example, Farshadfar et al. [82] observed that wheat genotypes produce improved soluble sugars at the grain filling stage than the pre-anthesis stage when exposed to drought stress.

Trehalose is highly water-soluble and chemically non-reactive in nature due to its non-reducing property. It embroils several metabolic processes to survive against abiotic stress through maintaining osmotic balance, stabilizes lipid membranes, functions as scavenging ROS, and protects protein synthetic machinery of plants [83]. Aldesuquy and Ghanem [84] observed that exogenous use of trehalose influences the intensification in peroxidase (POD), ascorbic acid oxidase (AAO), and phenylalanine ammonialyase (PAL) actions to shrink the drought-induced oxidative stress in wheat at the grain filling stage. Application of trehalose and maltose enhance the accumulation of total soluble sugars, proline, flavonoids, amino acids, and phenolic that lead to an increase in the survival ability of wheat against drought stress [85]. Besides, the accumulation of maltose during the grain filling stage of wheat facilitates the activity of PAL and polyphenol oxidase (PPO) for diminishing the abiotic stress-induced oxidative damage in plants [85]. Luo et al. [86] observed that high temperature (40 °C)-induced oxidative injury of wheat seedlings was reduced via pretreatment of trehalose. They detected that trehalose triggers the enhancement of β-carotene content, non-photochemical quenching, and amount of

deep oxidation of xanthophylls cycle pigments through decreasing chlorophyll content in wheat seedlings.

Sugar alcohols are composed of a considerable percentage of all assimilated $CO_2$, which act as predators of abiotic stress-induced oxygen radicals [87]. In general, sugar alcohols (polyols) comprise mainly mannitol, inositol, sorbitol, glycerol, arabitol, maltitol, and D-ononitol. The accumulation of mannitol is activated when the plants are subjected to environmental stress. It provides a steadiness of ROS-scavenging and macro-molecules [88]. Hameed and Iqbal [89] reported that treatment of wheat seed with mannose and mannitol enhances reducing sugar content in wheat leaves that provide osmotic adjustment in response to drought stress. The decreased lipid peroxidation activity and increased detoxification of ROS by catalase and ascorbate peroxidase are observed in roots of wheat plants after exogenous mannitol application under salt stress [90]. Inositol (myo-inositol) and its derivates such as pinitol, ononitol, and galactinol accrue in plants and accomplish numerous roles comprising osmoprotection [91]. They are not direct ROS scavengers but also perform as initial substrates for a possible biosynthesis of ascorbic acid for quenching ROS [92].

### 2.2. Production of Antioxidants

Plants have several self-protecting mechanisms to minimize plant cell damage from the effect of ROS molecules. The degree of oxidative damage depends on the amount of ROS production and activities of antioxidant components in the plant cell [38]. Among these physiological mechanisms, the antioxidant protection system is the most important one that involves both enzymatic and non-enzymatic components to prevent ROS aggregation [93]. The enzymatic mechanisms involve SOD, CAT, POD, ascorbate peroxidase (APX), monodehydro ascorbate reductase (MDHAR), DHAR, glutathione reductase (GR), glutathione-s-transferase (GST), and glutathione peroxidase (GPX) and directly scavenge ROS [94]. The non-enzymatic mechanisms such as glutathione (GSH), ascorbate (AsA), tocopherol, carotenoids, alkaloids, non-protein amino acid, and phenolic help to maintain the integrity of the photosynthetic cell membranes under abiotic stress-induced oxidative stress [44]. The tolerant wheat genotypes could mitigate the higher ROS activities through the expression of genes encoding SOD, CAT, APX, GPX, and GST enzymes under abiotic stress [95]. The SOD acts as a front defense mechanism against an increased amount of ROS. It is linked to convert $O_2^{\bullet-}$ into $H_2O_2$ and later into $O_2^{\bullet-}$ in the plastome, mitochondrion, cytoplasm, and peroxisome, whereas POD directly scavenges $H_2O_2$. The SOD is involved in post-translational modification and is reported to play a key role in drought tolerance [96]. In addition to detoxification via the tripeptide glutathione, GST isoforms may also act as glutathione peroxidases and thus are considered as integral parts of oxidative stress responses. Sairam et al. [97] found that the increased activity of SOD in salt-tolerant wheat genotype provides tolerance to salt stress compared to sensitive genotype. Likewise, Hameed et al. [98] stated the higher activity of SOD in tolerant wheat plants that assist in quenching $H_2O_2$ under heat stress. CAT enzyme mainly works to minimize the detrimental effects of oxidative damage through detoxification of $H_2O_2$ in the mitochondrion and micro-body [99]. Hasanuzzaman et al. [44] revealed that foliar treatment of $N_2O$ diminishes the heat-induced oxidative damage in seedlings of wheat through the increased activity of CAT antioxidant defense system. Similarly, Qiu et al. [100] observed that jasmonic acids (JAs)-treated seedlings of wheat showed a higher tolerance against salinity through increased activity of CAT. The higher activity of GPX provides an antioxidant defense in wheat seedlings upon exposure to short-term heat stress [101]. Ratnayaka et al. [102] demonstrated that the activity of GR or the APX was higher to protect cellular disruption from photo-oxidation by ROS during the recovery period of water stress conditions. Another earlier study confirmed that APX is acting as the significant antioxidant under water stress for scavenging $H_2O_2$ and superoxide radicals in the chloroplast of plant cells [103]. In wheat cultivars, APX activity mainly depends on the leaf developmental stage and severity of the drought. Under mild drought, the higher APX

activity was found in the leaves but under severe drought stress, APX activity was lower due to the increased malonic MDA production [104]. The enzymatic activity of GR plays a significant role in plant response to abiotic stresses through preserving decreased levels of glutathione (GSH), ascorbate (AsA) pools, and GSH/oxidized-GSH (GSSG) ratio than solely GSH content in plant cells [105].

Water-soluble phytohormone ascorbic acid (AA) [103] and lipid-soluble $\alpha$-tocopherol act as antioxidants to protect plants from abiotic-stress induced oxidative damage [104]. Besides, four enzymes, namely, dehydroascorbate, monodehydroascorbate and glutathione reductase, and ascorbate peroxidase are connected to the ascorbate-glutathione sequence for quenching of $O_2^{\bullet-}$ and $H_2O_2$ [105]. The above evidence was also confirmed by Farouk [106], who reported that decreased level of leaf senescence was found in the higher activity of AA and $\alpha$-tocopherol in several wheat genotypes through reducing the production of $H_2O_2$ under salt stress.

Carotenes were found essential components during the antioxidant defense system of plants, although they are very sensitive to oxidative stress [107]. The $\beta$-carotene is existed by bounding with the main complex of photosystem-I and photosystem-II in the chloroplasts of all green plants, which plays a significant role to protect photosynthetic tissue from oxidative damage through direct scavenging of triplet chlorophyll and inhibits the production of singlet oxygen [108]. The drought-tolerant wheat genotypes can minimize oxidative damage of $O_2^-$ by maintaining the increased level of carotenoids under drought stress [109]. Flavonoids are secondary plant metabolites that help to protect the plant cell from the harmful consequence of oxidative damage under environmental stress. Ma et al. [110] stated that the amount of flavonoids concentration is enhanced in wheat leaves due to higher expression of flavonoid biosynthetic genes under drought stress.

*2.3. Phytohormones*

Phytohormones (PHs) are chemical substances that are naturally produced at very low concentrations in different parts of plants. Although the major roles of PHs are improving the growth and development of plants, while PHs could also facilitate plant acclimatization to the effect of hostile environments [111]. PHs can influence the physio-biological developments of plants either by endogenous synthesis or exogenous application for abiotic stress tolerance in plants [112]. The major phytohormones are indole acetic acid (IAA), gibberellic acid (GA), indole-3-butyric acid (IBA), cytokinin (CK), abscisic acid (ABA), ethylene (ET), and salicylic acid (SA) [113,114].

Among these PHs, IAA is important for manipulating the development of lateral roots under drought and salinity. The IAA and IBA are the major forms of natural auxin that regulate plant growth and development such as cell division, cell elongation, differentiation, and apical domination [115]. Generally, seed germination is hampered as a consequence of soil salinity, while seeds treated with IAA or nepthalic acetic acid (NAA) showed the enhancement of germination under salinity stress [116]. The IAA-treated wheat seedling can avoid the harmful effects of soil salinity and has maintained the seedling growth. Sakhabutdinova et al. [117] stated that the amount of IAA production in the plant roots is gradually decreased in response to the intensity of salt stress. Similarly, Iqbal and Ashraf [118] found that salinity-induced oxidative damage was reduced in the wheat plant after seed treated with IAA. Besides, they also reported ionic homeostasis and induction of SA-biosynthesis occurred in the seed-treated plant. Akbari et al. [119] narrated that exogenous IAA application enhanced the length, fresh and dry weight of hypocotyls of three wheat genotypes under salinity stress.

Gibberellic acid (GA) plays an important role in seed germination, development of leaf and stem, lateral root growth, and blossoming [120]. It also acts to reduce lipid peroxidations induced by free radicals [121]. Plants rapidly synthesized GA under stressful conditions. Nayyar et al. [122] observed the highest germination of wheat seed by the treatment with 20 mg/l $GA_3$. Moreover, Kumar and Singh [123] reported that seed germination, plant growth, and grain yield of wheat were relatively higher due to $GA_3$

application than non-treated wheat under saline conditions. The wheat plant showed an increased level of germination and better plant development due to GAs application through enhancing antioxidant enzyme activity under salinity stress [124]. Similarly, Iqbal and Ashraf [125] described that effectual uptake and ion-partitioning within the wheat plant were higher due to $GA_3$ application that leads to increased growth and regulates the plant's metabolism under normal and stressed conditions. Tabatabei [126] also depicted that the exogenous application of $GA_3$ improved the tolerance of plants against salt stress through enhancing germination ability, and wheat plant growth by up-regulating antioxidant enzymes. The grain weight and grain quality of wheat were increased through enhancing the leaf area with photosynthetic pigments after seed treating with $GA_3$ under salt-stressed conditions [127]. Other phytohormones are involved in the developmental process of plants [128]. Besides these roles, CKs are also involved in stress tolerance in plants [129].

Abscisic acid (ABA) is considered a growth-limiting phytohormone and omnipresent in all types of flowering plants. ABA is normally known as the growth regulator that acts as a signaling molecule in plants during various environmental stresses [130]. It also has a dynamic role in different physiochemical activities of plants including maintenance of seed dormancy and root architectures, flowering, stomatal opening, embryo morphogenesis, storage proteins, lipids synthesis, and grain filling [131]. Abscisic acid is naturally generated and aggregated at low levels in the chloroplast. In general, stomata opening depends on the light but ABA and high $CO_2$ concentration influence partial or complete stomatal closure [132]. Moreover, Keskin et al. [130] depicted that the ABA-treated wheat plant activated *MAPK4*-like, *TIP1,* and *GLP1* genes that help to adapt wheat plants to stressful environments. The production of ABA is enhanced incrementally and helps plants against various stresses including water deficit condition, extreme temperature, and salt stress [131]. ABA biosynthesis is generally occurred in roots and then transferred to leaves for controlling of stomata opening through guard cell and protect from water loss during drought stress [133]. Moreover, Bano et al. [134] described that exogenous ABA application increases wheat tolerance to drought stress and protects plants from drought-induced oxidative damage through enhancing antioxidant defense mechanism. The ABA synthesis in the wheat plant serves as a promoter for root growth which has a positive correlation with the yield of the plant under drought stress [135]. Enhanced synthesis and aggregation of proline and ABA in wheat seedlings promote better growth and yield under salinity stress [136,137]. Ethylene (ET) is recognized as a growth regulator during various stressful conditions. An earlier study observed that biosynthesis of ET in plant cells increases abiotic stress tolerance of plants such as uneven temperatures, salt, and drought stress [138]. Narayana et al. [139] found that ethylene synthesis was enhanced in wheat leaves under water-stressed conditions. Furthermore, Young et al. [140] reported the onset of leaf senescence of cereals by ethylene biosynthesis under drought stress. Salicylic acid (SA) is an endogenous growth regulator which involves in the physiochemical processes of plants [141]. It also participates to protect plants from numerous stresses through the modulation of antioxidative enzyme activities [117].

### 2.4. Role of Plant Growth-Promoting Microorganisms (PGPM) during Abiotic Stresses in Plants

The rhizosphere is the rich niche of taxonomically diverse groups of PGPM. The release of root exudates by plants is an indispensable factor for microbial colonization in the rhizosphere [141]. Chemotactic development of microorganisms toward the root exudates assumes the job of hauling power for the microbial networks to colonize the roots. The rhizosphere microorganisms act as indispensable protagonists in nutrition, disease protection, tolerance to stress, and the growth of plants by various direct and indirect mechanisms [142]. While using the rhizosphere-microenvironment around plant roots, the plant growth-promoting rhizobacteria (PGPR) may go about as biofertilizers, phyto-stimulators, or biocontrol operators relying on their innate abilities, method of collaboration, and serious endurance conditions [143]. A large body of literature revealed

that rhizosphere microorganisms defend plants from different stresses [144]. The discovery of these beneficial microorganisms and their application as novel biologicals significantly reduce the hostile effects of numerous environmental stresses. Some of the plant-associated beneficial fungi and bacteria have been used as biostimulants, biopesticides, and also biofertilizers [13,145].

The vesicular-arbuscular mycorrhiza (VAM) plays important role in enhancing the adaptability of plants against stress and promoting the growth of plants. These beneficial fungi rather may occupy host plants on the surface (ectomycorrhizal) or they may frame endosymbiotic affiliations (VAM). These organisms structure broad systems administration of fine hyphae, subsequently expanding generally supplement take-up by the roots.

The root contagious endophyte *Piriformospora indica* instigates salt resilience in wheat grain and heat resistance in Chinese cabbage by expanding the degrees of cell reinforcements and improving cells by different mechanisms. Toward one side, microorganisms prompt neighborhood or foundational stress lightening reaction systems in plants (Figure 1) to continue under hostile environmental conditions, while at the opposite end, they help plants to keep up their development and advancement through obsession, activation, and additionally synthesis of supplements, hormones, and natural phyto-stimulant mixes. Such multifaceted activity of microorganisms or their networks makes them solid, suitable, and essential choices for environmental stress alleviation systems in plants [146].

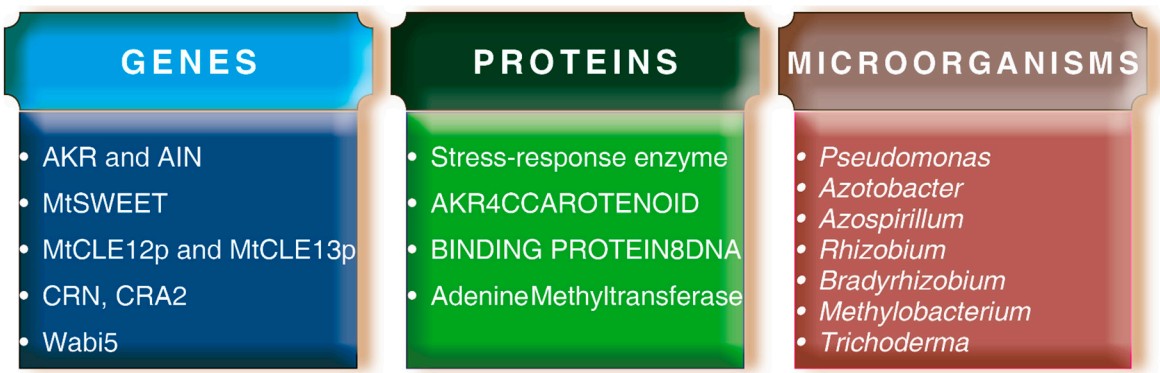

**Figure 1.** Elements involved in fitness and molecular and genetic improvement of the crop plants.

Enhancement of plants' tolerance to unfriendly environmental conditions such as salt stress, water deficit stress, heavy metals, life-threatening temperature, and submerging by the application of various beneficial microorganisms has been reviewed by Chatterjee et al. [147]. Mechanisms involved in the adaptability against various stresses in plants by microorganisms include altering the content of phytohormones, changes in root and shoot morphology, increase in relative water content, biosynthesis of scavenging enzymes, accumulation of osmoprotectants, and enrichment of nutrient uptake by the plants [13,146]. Genes involved in these mechanisms of functions of beneficial microorganisms have been discovered and some of the discovered genes have been introduced in crop plants. Transgenic plants expressing the microbial stress tolerance genes showed higher tolerance to abiotic stresses [147]. These findings have opened an array of opportunities for the application of beneficial microorganisms in both agriculture and biotechnology [13,147]. A better understanding of molecular cross-talk between these microorganisms and plants would help to exploit them to alter plant metabolism and provide resistance to abiotic stresses [144].

### 2.5. Breeding, Biotechnology and Genetic Engineering Approaches for Stress Tolerance in Plants

Traditional plant breeding approaches supplemented with marker-assisted selection and genetic editing, as well as high-throughput phenotyping techniques, are exploited to speed up the breeding for the desired genotypes. Biochemical and genetic bases for the enrichment of the grain of modern cereal crop cultivars with micronutrients, oils, phenolics,

and other compounds are discussed, and certain cases of contributions to special health-improving diets are summarized [148]. *Triticeae,* an economically important tribe that includes major crop genera, such as wheat, barley, and rye, has been extremely successful in taking advantage of this speciation mechanism. The *Triticeae* tribe within the grass family *Poaceae* (*Gramineae*) includes about 360 species, from which only 80 species are diploid, being the majority allopolyploids derived from interspecific and intergeneric hybrids [149]. Among several different genome-modifying tools, the CRISPR/Cas9 system is the recent and widely used genome modification tool because it is simple and highly efficient technology. The CRISPR/CAS9 system along with its variants has immense potential to develop new wheat varieties with higher yield potential [150].

### 2.5.1. Conventional Breeding

Numerous endeavors have been done to distinguish pressure lenient germplasm and to investigate the components of stress resistance at the genomic level. Breeding for various environmental stresses adaptability in wheat plants has long been practiced as a classical and safe tool for wheat improvement. Many abiotic stress-tolerant wheat varieties have been developed by conventional breeding. However, conventional breeding is a time consuming and labor-intensive approach. Due to its hexaploid nature, wheat is complicated compared to diploid rice and maize. Hereditary fluctuation for stress resistance is being screened among accessible genotypes of oat crop species and their crop wild relatives (CWR), with the point of working up a thorough genetic stock to be utilized in rearing projects. Approaches directly involved in conventional breeding are discussed in Figure 2.

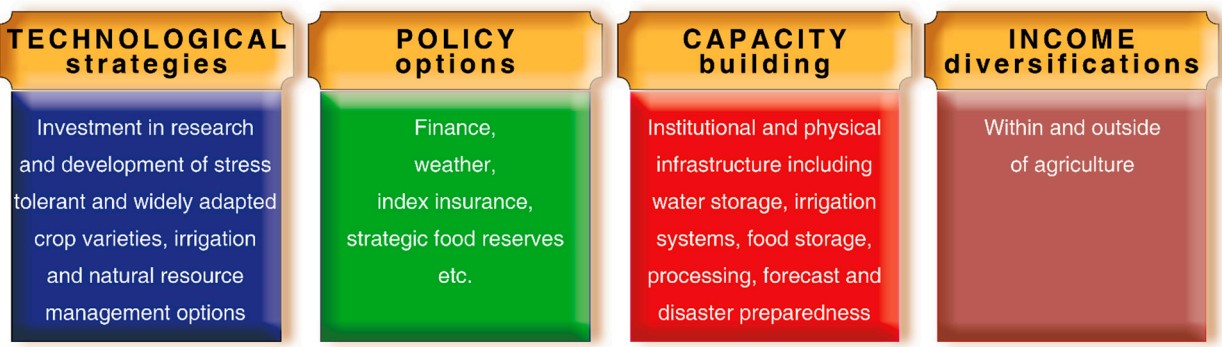

**Figure 2.** Approaches involved in conventional breeding in traditional farming.

Reproducing for transformation to ecological anxieties needs to confront expanding vulnerability in the event of stress-creating atmosphere occasions [151]. An essential inquiry is accordingly: 'Which reproducing and the board procedures can be imagined to guarantee great harvest execution in an unpleasant domain commanded by the vulnerability.' The accessibility of a huge pool of cultivars may not be sufficient to face such vulnerability. Expanded harvest hereditary heterogeneity, either by utilizing cultivar blends or heterogeneous transformative populations like composite cross populations, may preferably adapt to ecological vulnerability over the utilization of a solitary and homogeneous cultivar. On account of developmental populations, dynamic variation has been reported to, for example, improve winter endurance or salt resilience in grain composite cross populations. Variety in wheat heading and flowering time might be viewed as a pressure evasion procedure [152] when developing and imitating subsets of a similar wheat population in various climatic conditions for a long time. Regardless of little exploration has been done on the connection between intercropping and abiotic stress resistance, proof of improved supplement or water use productivity in grain vegetable intercrops proposes that species assorted variety can assume a significant job additionally in buffering natural variety and vulnerability [153].

Germplasm Selection

Genotype selection under a target stress condition could provide a better result. Genotypes may vary in their performance in different abiotic stresses. In general, a single genotype might provide good yield under a specific stressful situation. In rare cases, a cultivar could show better yielding potential under more than one stress. Those genotypes bear great importance for combined stress. For sustaining under the stressed situation, a genotype often compromised with final production. Thus, yield potentiality would be the main concern while selecting abiotic stress-tolerant/resistant cultivars. Genotypes that produce better yield under stress conditions have underlying physiological and molecular mechanisms to combat stress. Sometimes secondary physio-morphological characters play a protagonist in conferring resistance to hostile environments [154]. So, it is imperative to consider the secondary traits along with the grain yield in selecting cultivar suitable for stress environment.

Testing Environment

The selection of the environment is a distinctive important component for resistance breeding against abiotic stress in plants. The testing environment should be similar to the cultivating natural environment. When selection and target environment are similar, indirect selection of genotypes might be beneficial [155]. The stress level should be imposed accordingly. A higher level of stress during testing could bring better resistant cultivar. Abiotic stress sometimes has a compound effect with other abiotic stress, such as heat stress often comes with drought stress. Prasanna [156] expressed that heat stress coupled with drought is becoming an increasing reality; whereas Chen et al. [157] suggested that two foremost abiotic stresses (i.e., drought and heat stress) harshly prohibited growth and development of plants lead to extensive loss of crop productivity worldwide. In such cases, facilities are needed to be developed for multiple abiotic stresses to select stress-resistant genotypes. Besides, genotype interaction with the target environment is also important for stress breeding approaches and can be affected by various abiotic stresses [158,159]. Genotype x environment interaction study might be the best breeding approach in selecting germplasm conferring stress resistance in crops including wheat [160]. This study under multiple environments of specific stress situations could bring better cultivar having improved resistance to stress in wheat.

Target Traits

Selection of germplasm based on both primary traits like yield and secondary traits which are significantly associated with yield potential could ensure genetic gain under stressful environment. Solely, yield-based selection could be sometimes misleading under abiotic stress conditions and would be less efficient due to low heritability under stress [161]. Besides these, the association of grain yield during hostile and non-hostile environment is not consistent [162]. In such a condition, one of the effective options is to go for indirect selection through secondary morpho-physiological traits that had a significant correlation with the yield at stress environment [163]. To date, various secondary traits have been considered other than grain yield for crop improvement under abiotic stresses including heat stress, drought stress, etc. [154]. However, a secondary trait needs to fulfill the criterion of being both genetically variable and heritable, fast to measure, and associated with yield under stress [164]. Grain yield along with specific secondary traits in multi-environment trials might be useful while focusing on genotype x environment interaction under abiotic stress tolerance [164]. Apart from the morpho-physiological traits, selection indices like stress susceptibility index, stress tolerance index, etc. could also provide important information in selecting stress-tolerant genotypes [165].

Heterosis Exploitation

Deploying heterosis to confer resistance to abiotic stress could be interesting in the wheat breeding program. Hybrids could perform better than pure lines or open pollinated

(OP) in expressing particular characters under stressful conditions. Best parent heterosis in hybrids was reported as consistent in field trials for yielding potential [166] for low levels at the grain filling stage [167]. Elevated drought resistance was concluded for hybrids [168] and can perform even better than parents for yield and yield attributing characters during water deficit stress [169]. The exploitation of heterosis for several root characters was reported in earlier findings on wheat [170]. However, heterosis could be utilized for abiotic stress breeding in wheat. There are landraces and wild relatives which are a rich source of abiotic stress resistance genes [171]. Those particular genes can be transferred in developing new crop cultivars through hybridization [172] which would show better resistance to abiotic stress.

### 2.5.2. Molecular Mechanisms
#### Genome-Wide Association (GWAS)

Various environmental stress tolerance in different plant species including wheat is a complex quantitative trait. To reveal the underlying molecular mechanism of such complex traits, scientists over the world are employing GWAS. It generally involves a huge panel of diversified germplasm to provide a recombination event-based high-resolution mapping of genes [173]. Though the panel size is a big limitation of this study, several studies were conducted by the researchers on different crops such as maize [174], in wheat [175]. At least 10 GWAS were carried out in the last five years and approximately 960 marker-trait associations (MTAs) were identified for different traits under heat stress [176]. A large number of MTAs have been recognized from GWAS for drought stress-related traits in wheat in recent studies [177]. Consequently, at least 46 candidate genes were detected for several traits [178]. In the case of salt tolerance, several MTAs were detected in wheat from GWAS were related to the function linked to the stress [179].

#### Quantitative Trait Loci (QTL) Mapping

Bi-parental population-based mapping, i.e., QTL mapping, is a recognized tool for detecting genomic regions associated with target traits. Both segregating and fixed population from two parents are being utilized in searching genomic regions through QTL mapping. This approach is a powerful statistical tool [180] and can also be used for abiotic stress resistance studies. In identifying the genomic region, QTL mapping strategies were successfully exploited in many studies for abiotic stress tolerance including drought [181]. Quite a huge quantity of QTLs was detected in various crops [182]. Large effect and stable QTL for traits associated with heat stress were identified in different studies [48]. The QTL for drought stress-related traits has also been detected in several studies [183]. The QTL for salinity tolerance was identified in more than 20 studies while reporting approximately 500 QTL [184,185].

#### Multi Parent Populations (MPP)

Multi-parent populations (MPP) involve more than two parents in developing mapping populations. This is a comparatively new approach in molecular breeding. There are limitations in using a single approach like GWAS and QTL mapping alone, and scientists are trying to deploy both approaches in the MPP. The different working groups successfully developed MPP in a few crops such as nested association mapping (NAM) in maize [186] and MPP advanced generation intercross (MAGIC) in wheat [186] and rice [187]. The MAGIC populations are already exploited in wheat to develop a linkage map [188]. The MAGIC population could also be developed and utilized for abiotic stress tolerance studies.

#### Backcross Populations (BC)

Backcross (BC) populations involve a cross among a recurring and a contributor parent. The target traits are carried by the donor parent. As BC carries mostly the recurrent parents' genetic constituents, it holds importance in transferring stress tolerant genes to the subsequent progenies. An elite cultivar with a target gene can be developed through

marker-assisted backcrossing [189]. This approach was successfully exploited in developing chromosomal segment substitution lines (CSSLs) as well as in identifying QTLs for a range of traits in many crops [190,191]. Backcross population is also employed in drought stress tolerance study [192]. This could also be extended for other abiotic stress tolerance studies.

Homozygous Immortal Populations

Recombinant inbred lines (RILs), near-isogenic lines (NILs), and doubled haploids (DH) are the fixed populations that are being utilized in genomic studies. Those populations had a homozygous genetic background, and generally, develop and deploy invalidating the previously detected QTLs for target traits. The RILs, NILs, and DH have been exploited in developing new cultivars and in genomic analysis in many crops including wheat [193] and a few of them were on abiotic stress tolerance studies. Those populations might be exploited explicitly in different environmental stress tolerance studies in wheat plants.

2.5.3. Biotechnological Tools for Various Environmental Stress Tolerance in Plants

Plant interactions toward different abiotic stresses and microorganism intervened pressure relief procedures in plants that have been concentrated on sound grounds of atomic, biochemical, physiological, and ultrastructural boundaries. Such examinations have been done using diverse omics approaches (genomics, metagenomics, met-transcriptomics, proteomics, met-proteomics, and metabolomics) (Figure 3) that reinforced our comprehension behind the instruments of microbial associations, quality falls and metabolic pathways, gathering and upgrade of different metabolites, proteins, catalysts and all over guideline of various qualities. Such examinations could yield dynamic information identified with joined reactions of plants to numerous burdens, and the equivalent is additionally appropriate with the normally related or falsely vaccinated microorganisms [194,195]. Omics analyses give new headings to extemporizing the current conventions in the field of plant–organism communications under pressure, and utilization of microorganisms and microbial metabolites for easing of various burdens experienced by plants. The general results are encouraging germination, prevalent food, improved capacity to battle unfavorable states of condition, and unrivaled yield in plants in light of the utilization of organism expounded particles. We firmly advocate that there is a need for prominent inside related to recognizable proof, characteristic portrayal, similarity index, convenient techniques, and effect of utilization of microorganisms for the relief of abiotic stresses in crop plants. We have to discover new microbial metabolites that are being delivered under focused on ecological conditions. Built-up confirmations exist to help the job of microorganism interceded plant associations in stress relief under various climatic and edaphic conditions. In any case, a more engaged omics-based exploration information age following coordinated methodologies incorporating genomics, metagenomics, proteomics, and metabolomics concentrates on explicit plant–microorganism abiotic stress framework will be expected to determine numerous realities behind exact systems of stress resistance/moderation in the harvest plants [154].

Tissue Culture

Tissue culture has efficiently been used for years in plant improvement. In recent decades, plant tissue culture-based techniques are being exploited in addressing various stress tolerance in plants [196]. In plant improvement and creating genetic diversity, tissue culture protocols are important such as somaclonal variation, in vitro selection, doubled haploids technology, wide hybridization, etc. [197–200]. The proteins related to LysM genes in wheat are classified into seven different groups. Synteny analysis suggested that the LysM family plays a very important role in the immunity of the wheat and other species [201].

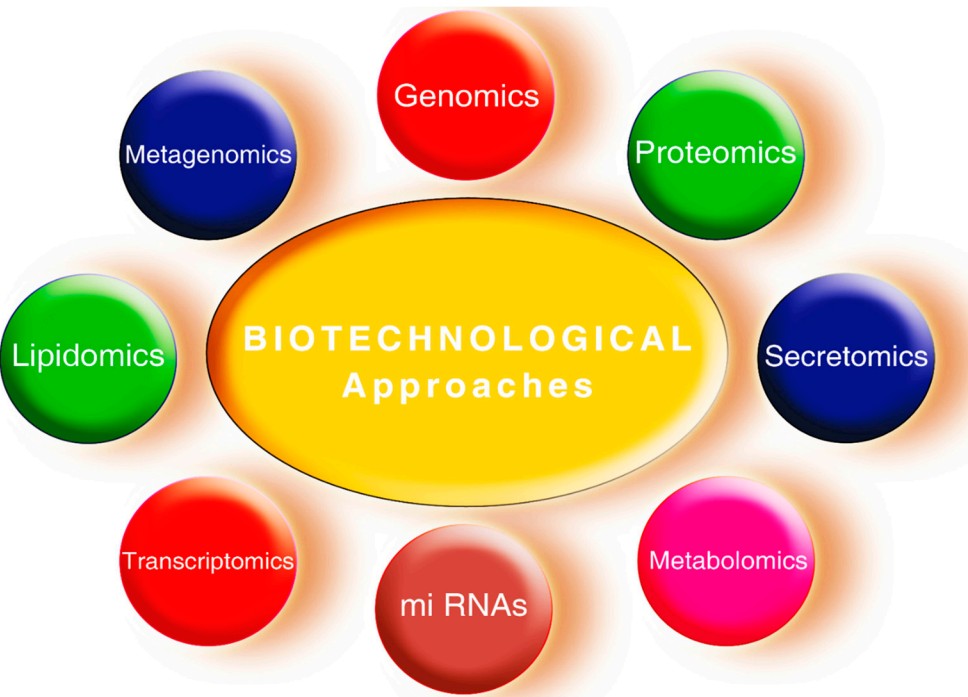

**Figure 3.** Different omics approaches for crop improvement.

Somaclonal Variation

Somaclonal variation is the genetic variability produced during cell culture in vitro [158], which is also known as tissue culture-induced variation [202]. Its exploitation in breeding crop plants has been reported [203], which can be used in manipulating genetic constituents for polygenic traits and can exhibit elevated resistance and improved yield [204]. This method is useful in abiotic stress tolerance breeding programs in crops [205]. The variation could be used to develop agronomically superior wheat cultivar [206] as well as in the improvement of drought tolerance in wheat [207].

In Vitro Selection

In vitro selection through tissue culture method has been used for both biotic and abiotic stress breeding [200]. So far, several studies have been reported on drought tolerance in different crops including wheat [205]. Resistant lines have been isolated for biotic stress tolerance in wheat [206]. Salinity and zinc tolerance have been attributed by in vitro selection [207]. It holds importance in classical breeding techniques [208] and can be used following somaclonal variation [200]. Stress tolerant lines from traditional as well as the transgenic breeding approach could undergo in vitro selection process and would be suitable for some abiotic stresses [200]. Plants susceptible to salinity stress can be turned tolerant by mutations in the responsive gene(s) [209]. Drought tolerant wheat lines could also be developed through in vitro screening method [210]. However, in vitro selection is found to be an effective technique for developing stress-tolerant plants [211] and can be used for different crops including wheat.

*2.6. Genetic Engineering*

2.6.1. Genetic Modification Approach

Genetic engineering strategies are efficiently using in crop improvement especially to answer various stresses. It is a rapid solution compared to the traditional breeding approaches and is in practice for crop improvement [212]. Various gene delivery methods such as *Agrobacterium*-mediated, as well as particle bombardment techniques, are generally used for engineering desired plant cells for the improvement of targeted traits [213]. A

huge number of literatures are obtainable on the improvement of various environmental stress tolerance in plants by genetic engineering. For example, scientists in Egypt have successfully developed drought-tolerant wheat cultivar through the transformation of '*HVAI1*' gene from barley to wheat [214]. More work is going on to develop genetically modified drought-tolerant wheat cultivar in China and other countries [215]. Genes from wheat *TNHX1* and *TVP1* transferred in *Arabidopsis* and the transgenic exhibited enhanced tolerance to both drought and salt stresses [216]. Genetic modifications of crop plants including wheat by recently developed CRISPR-Cas genome editing technology have recently been reviewed [13].

### 2.6.2. Gene Expression and Functional Genomics

The International Wheat Genome Sequencing Consortium (IWGSC) released a fully annotated reference genome of wheat [217]. It hugely facilitates the molecular breeding of wheat for the development of various hostile environmental tolerance. In both forward and reverse genetics, the gene function needs to be verified through the overexpression of genes that can provide elevated abiotic stress tolerance in crops [218]. The efficiency of a transgenic approach depends on gene expression analysis [200]. Comprehensive gene expression and functional analysis will involve gene function, molecular and biochemical mechanism, and signaling networks activated during stresses imposed. A complex signaling network was reported when reviewed genes functional analysis for transcription factors [219]. Transcription factors are the major transgenes that regulate the expression of other genes under stress [218]. In transgenic wheat, drought tolerance is improved by utilizing stress-inducible promoter rd29A with *DREB1* from Arabidopsis [220]. Numbers of studies are underway worldwide; several hundreds of patents have been made for stress tolerance [221]. Although hundreds of thousands of candidate genes were identified for abiotic stress tolerance, their practical utilization in developing commercial variety is very slow [218]. The application of the revolutionary CRISPR-Cas genome editing technology should accelerate the development of new wheat varieties tolerant to multiple abiotic stresses [13,32]. Bhowmik et al. [222] have developed a convenient method for mutagenesis of wheat using CRISPR-Cas technology.

### 2.6.3. Molecular Marker-Assisted Breeding

Marker-assisted selection (MAS) using molecular marker can speed-up the selection process, as it is two times speedy than traditional breeding for traits like stress tolerance, yield, etc. [223]. It is already been deploying in abiotic stress tolerance breeding [224]. It is cost-effective and can trigger long-term development in stress breeding [225]. It is more effective for different complex traits controlled by multiple genes [200]. Under drought stress, three QTLs were identified in wheat [226], while in other studies, 13 QTLs were detected in the F4 population of drought susceptible and tolerant cross [227]. Salinity tolerant wheat cultivar can be developed using this tool [225], and QTLs have been detected related to salt tolerance [228]. Several QTLs have already been recognized [229] which can be used in waterlogging tolerance breeding in wheat [230], whereas simple sequence repeats (SSR) markers have been detected and could be exploited in developing AL tolerant cultivar by MAS [231].

### 2.6.4. Gene Pyramiding Assisted by MAS

Gene pyramiding is the strategy of accumulating more than one gene of target traits into a single cultivar holds great importance in plant breeding. Achieving sustainable resistance to abiotic and biotic stress is possible through pyramiding different tolerant genes [200]. In general, it involves gathering two or sometimes more genes into an elite variety to enhance the characters' expression [232]. Pyramiding promising target traits can conveniently be done involving MAS [233]. It has been successfully used for biotic stress resistance in common bean and soybean [234] and can also be used for abiotic stress tolerance in wheat.

### 2.7. Double Haploid Breeding

Doubled haploid (DH) breeding technology refers to the development of haploid plants through anther culture followed by chromosome doubling. The speedy development of homozygous lines can be achieved by this technology. Moreover, the cost and time required for cultivar development are much less than the traditional approach [235]. Those lines are fixed populations that can be treated as RILs for mapping and also useful in a breeding program [236]. In wheat, better tolerance under a water-limited environment was observed in double haploid transgenics [237] and also reported that this DH can be used in functional genomics studies in wheat. Besides, DH populations will provide wheat breeders several options including cultivar development with better tolerant genes, transferred from wild ones [238]. However, DH in coupled with MAS would be more effective for selection for desirable traits [200].

### 2.8. Wide Hybridization

Hybridization between cultivated species and their wild relatives (CWR) is termed wide hybridization. It can be utilized in broadening the genetic constituents of the pool might have genes to confer tolerance to biotic and abiotic stresses [239]. Many wheat genotypes have already been developed through wide hybridization which showed better tolerance to biotic and abiotic stresses [240]. Somatic hybridization through protoplast fusion is another strategy that has been exploited by plant breeders in crops for the improvement in performance [241]. In wheat, multiple attempts have been made for interspecific and intergeneric hybrids to date, though the molecular events are unknown in regenerated plants but could be useful in functional genomics and stress tolerance research [242]. However, utilization of this technique has also been reported for abiotic stress tolerance in different crops including wheat [243].

### 3. Nano-Technology for Plants Tolerance to Abiotic Stress

Nanotechnology is considered an emerging field to survive against various environmental stresses in wheat and achieving assured productivity and to address the present-day challenges associated with food security as well as with the food system [244]. Nanotechnology has the potential to change the production system through different devices to precisely detect and measure soil fertility status, soil moisture level, and nutrients level in plants along with temperature and disease pests in wheat [22]. These nano-tools also help in assessing real-time crop growth and deliver essential information for precision farming. Nano-sensors and other sensing devices provide important information on the optimal time for sowing and harvesting wheat and can provide valuable information for the timely application of various agrochemicals [245]. Numerous studies indicated that nanotechnology performs a significant role in mitigating hostile environment-induced variations in wheat. Further, multi-walled-carbon nanotube (MWCNT) performs a significant role to influence seed germination and regulate plant growth in wheat [246]. This tool helps in increasing essential nutrient (Ca and Fe) and water uptake efficiency which could boost seed germination and plant growth in wheat. Titanium dioxide nanoparticles (NPs) ($nTiO_2$) are an effective and useful nutrient source for plants to develop biomass production by improving nitrogen assimilation in wheat plants [247]. It has been observed that the use of nano titanium dioxide at the optimum level stimulates the growth of shoots and roots in wheat [248]. Drought is considered as severe environmental stress in wheat production across the globe and a key constraint on wheat productivity with an important impact on growth [249]. It is observed that the application of analcite NPs to the soil with an optimal dosage improved germination, growth, and development of wheat under heat and drought-prone environment. Silicon (Si) is considered a beneficial element in the plant during hostile environments. Si is naturally not considered as an essential nutrient element in plants but its high accumulation rate in monocots (i.e., wheat) through the root system improves drought tolerance through enhancing hydraulic conductance of root [250]. It is well recognized that soil salinity is the most significant abiotic stress in wheat production.

Poor germination along with a reduction in crop stand is the prime result of soil salinity which negatively affects crop yield [251]. The effect of salinity has been documented at the early vegetative stage, varies from the lessening in germination percentage to inconsistent uptake of different nutrient elements due to inhibition in proper root growth in wheat [252]. It is also stated that soil application of Green Cu-NPs (25 and 50 mg kg$^{-1}$) effectively reduces the antioxidant stress Cu translocation vis-à-vis significantly enhances the plant growth and yield of winter wheat [253–257]. However, the utilization of NPs not only boosts plant growth but also enhances germination performance in wheat under salinity stress [254]. It is informed that the seed priming with Ag-NPs can improve salt stress tolerance in wheat by lessening the oxidative damage through modifying antioxidant enzymatic activities. Moreover, it is observed that the application of Ag-NPs in different doses can reduce various salt stress-induced symptoms in wheat [258].

## 4. Agronomic Approaches for Abiotic Stress Management

Abiotic stresses in wheat are globally an increasing concern and its tremendous impact on crop production is increasing due to global warming and climate change [259]. Various approaches to abiotic stress mitigation strategies in the wheat production system escalated by global warming and climate change have been documented in different research articles [260]. Agronomic management like adjustment of sowing time, efficient nutrients and irrigation management, and selection of suitable climate-smart technologies can not only alleviate the various abiotic stresses but also increase the yield and production use efficiencies in wheat.

### 4.1. Adjustment of Sowing Time

Optimum sowing time is the most significant influence which determines the yield of crops. Delay in sowing of wheat beyond the optimum sowing window might have a negative effect on crop yield [261]. There is an optimum sowing date exists for each specific location, and yield reduction is observed when sown before and after this date [262]. Agro-metrological factors like air temperature, solar radiation, and seasonal distribution of rainfall also play a significant role to determine the yield [263]. Selection of optimum sowing time for wheat is one of the most crucial agronomic managements to ensure assured germination of the seed, steady seedling establishment, and yield maximization of wheat and this is directly linked with the directive of various biotic and abiotic stress [264]. Sowing of wheat crop beyond the optimum sowing window may lead to suboptimal production, even with the congenial weather condition. Delay in sowing of winter wheat can consequence in poor seed germination, vegetative growth, and lower tillering capacity due to uncongenial growing conditions [265]. Shah et al. [264] reported that grain yield of winter wheat in China declined by 1% for delaying each day from optimum sowing date may be due to inhibition of crop growth and yield attributing traits. Panhwar et al. [266] also stated that a delay in sowing of wheat of about one month from the optimum sowing time may reduce 34% grain yield. Moreover, the lower yield was largely attributed to the grain-filling phase coinciding with the diminishing temperature and radiation levels, thereby causing a decline in grain weight [267]. In the wheat-growing belt of north-western India, delayed sowing of wheat due to late harvesting of rainy season rice, imperilled the wheat to heat stress during grain developmental stage. This oddity of the situation is causing high-temperature stress in wheat followed by a significant reduction in yield [268]. Significant yield enhancement dew to early sowing of winter wheat has been reported by many researchers [269]. These enhancements might be because of good photosynthates accumulated in leaves and transferred subsequently to economic parts (grain) [270,271]. The delayed introduction might is associated with a significant decline in yield attributes [272]. These results seem to suggest that the environmental changes associated with different sowing dates might have modifying effects on the plant process, hence, the variation in the yield and yield components [273].

In practical field conditions, it is very different to recommend a universal sowing time for wheat cultivating in the diverged wheat growing region in the world. The selection of optimum sowing window for a location largely depends on the agro-climatic condition, existing cropping systems, and finally farmers' ability to adapt the modified agronomic practices. Crop simulation models that assess the complex phenomena amongst the plant-soil-atmosphere continuum and farmers' management practices, might be a very useful tool for determining the suitable cropping window for achieving the maximum yield vis-à-vis alleviating the abiotic stress in wheat [273,274]. APSIM, a dynamic cropping system simulation model, could be a potential support tool for the selection of suitable cropping window, particularly rice-wheat systems of South-Asia [275,276]. Bai and Tao [275] also successfully simulated the rice–wheat rotation system using APSIM-Oryza and wheat model to optimize crop cultivar and sowing time, water and nitrogen use efficiency, and environmental impact.

### *4.2. Nutrients Management*

Efficient nutrient management strategies through judicious use of organic and inorganic plant nutrients are crucial mechanisms in the improvement of the abiotic stress tolerance in wheat crops. Different macro and micronutrients play their specific role to regulate the stress responses in the wheat plant [277,278]. Source, time, methods, and amount of the plant nutrient application are the key factors behind the effectiveness of plant nutrients to mitigate the abiotic stress in wheat.

### 4.2.1. Organic Nutrients Management

In wheat cultivation, the use of organic manures is more profitable than inorganic ones under high-temperature stress. Volatilization loss of nitrogenous fertilizers is less when applied in the organic form under high-temperature stress. Besides, organic fertilizers increase yield in wheat farming under temperature stress [279]. Straw mulching is also considered an important agronomic input that helps to ameliorate moisture stress in wheat plants under high temperature [275]. Straw mulches are used as a potential tool to reduce soil moisture evaporation and increase the infiltration rate in wheat. Application of organic plant nutrients can augment the crop and water use efficiency of wheat under the deficit supply of irrigation water [280]. It has also been exhibited in various articles that straw mulching reduces the soil temperature and enhances seed germination coupled with improving plant growth in wheat [281]. It has been observed that most of the bulky organic manures help in the moisture conservation process under moisture stress conditions in wheat. Organic fertilizers could increase the soil moisture-holding capacity under water stress conditions in wheat [282], and thereby improves water retention capacity in soil micro and macropores and ultimately enhanced productivity in wheat during drought stress. Besides, several types of research portrayed that the use of organic manures in water stress condition not only enhance water use efficiency in plants but also increase N availability in the root zone in wheat [283].

### 4.2.2. Inorganic Nutrients Management

Nitrogenous fertilizers are the greatest significant plant nutrients for the growth and yield of the wheat crop. Nitrogen (N) acts as a vital role in governing the numerous physical, physiochemical processes in plants [284]. It is reported that effective N-nutrition management can potentially alleviate the drought stress in wheat by sustaining different metabolic processes [285]. Proper supply of nitrogenous fertilizers could accelerate plant growth, improve water use efficiency (WUE), and ultimately lessen the negative impact of drought stress [286]. N fertilization and irrigation management possess a synergistic relationship in alleviating the drought stress in wheat. Judicious application of inorganic N fertilizers and irrigation water not only reduces the drought stress but also significantly increases the yield and WUE of wheat [283].

The application of potassium into wheat can boost resistance against various abiotic stresses [276,277]. The use of K fertilizer improves different physiological traits related to photosynthesis and enzymatic activities, which eventually accelerates the development and productivity of wheat under abiotic stressed conditions [284]. However, the application of K fertilizer in wheat plants exposed under abiotic stress can increase physiological efficiency by lessening the uptake of toxic elements and improved drought resistance at various growth stages [285]. Moreover, the maximum improvement in plant physiological traits and nutrient uptake was attained when potassium was applied at the grain developmental stage [286]. It has been revealed that the accumulation of K fertilizer in wheat plants can regulate internal water balance and control various physiological traits related to water stress and yield attributes [45]. The use of phosphatic fertilizers under abiotic stress conditions also acts a vital role in increasing root biomass and helps in improving water-extracting capacity. Additionally, P fertilizers mainly amplified leaf water content and improve other physiological attributes associated with a photosynthetic rate under abiotic stress conditions [287]. Therefore, the use of P fertilizers can build up drought tolerance through physio-biochemical adjustments in wheat plants. It has been confirmed that the application of zinc (Zn) during hostile environmental conditions can moderate the detrimental activity of free radicals in the wheat plant. Besides, the application of Zn fertilizers in wheat can diminish the yield reduction under water stress conditions. Copper (Cu) performs a vital role in strengthening plant cell wall and external application Cu assists to survive the wheat under various abiotic stress [288]. However, boron (B) is also an important micronutrient in alleviating water stress in the wheat plant [254,255]. The application of boron in plants mitigates the water stress by controlling different physiological traits in plants and enhances growth attributes in wheat [289].

*4.3. Irrigation Management*

Irrigation management in the wheat production system is one of the crucial approaches to alleviate the osmotic vis-à-vis heat stress [290–293]. Judicious application of irrigation water is not only a driving factor for augmenting the yield but also mitigating the various abiotic stresses. Wheat is considered as one of the water stress-sensitive crops and sometimes limited supply of irrigation water cannot be able to encounter the full water demand of the crop [294]. Reproductive and grain formation states are the most critical growth stages of wheat for moisture stress. It was reported that wheat yield might be reduced up to 30% and 92% depending on the severity of moisture stress at post-anthesis and grain formation stages, respectively [295]. The rooting system of wheat plays a crucial role in regulating the moisture uptake pattern and nutrient uptake, partitioning of the photosynthates are largely depends on the interrelationship between soil moisture and root system [296]. It has been reported that moisture stress at critical growth stages of wheat significantly diminishes nutrient uptake vis-à-vis curbs root growth and development [294]. Modified application of irrigation water like ridge-furrow planting system in combination with plastic film mulching significantly improve moisture and nutrient uptake followed by reduced drought stress [297]. Efficient nutrient management especially the external supply of nitrogenous fertilizer possesses a close relationship with irrigation management to regulate the crop stress vis-à-vis yield level of wheat [298]. Thus, optimization of irrigation and nitrogen management can significantly enhance photosynthesis ability and yield potentiality and also help in reducing the different abiotic stresses in wheat [266]. Si et al. [299] reported that combined application of 240 kg N ha$^{-1}$ along 40 mm water per irrigation was optimally maximizing the yield, WUE, and reducing the terminal moisture stress for drip-irrigated winter wheat. Mitigation of moisture stress and improvement of WUE in the wheat crop in rainfed condition by adoption of the drip irrigation system was also observed [300,301]. Exogenous application of osmotic balancing solutions like glycine betaine and growth regulations like methyl jasmonates, abscisic acid, and salicylic acid can effectively improve the WUE and drought stress by controlling photosynthetic efficacy and antioxidative capability of winter wheat under limited irrigation conditions [302]. A

recent study conducted by Selim et al. [303] revealed that the application of magnetic water (water was developed through artificial magnetization process) meaningfully improved the growth and yield of winter wheat grown udder drought stressed condition. Application of modern remote sensing tools like satellite imagery, microwave remote sensors for assessment of drought stress of wheat from a large area is being used increasingly [304]. Modern crop simulation models like APSIM may also be a handy decision support tool for scheduling precious irrigation water under a rainfed and irrigated wheat cultivation system [305].

### 4.4. Climate Smart Technology

Climate change and associated increased $CO_2$ concentration in the atmosphere expressively stimulate the growth and development lead to improve the yield of wheat under different growing ecologies. A vivid understanding is essential to alleviate the adverse impact of the climate-changing on rice-wheat cropping systems (RWCS). Being the principal crop grown in the South-Asian countries, it is essential to learn the relationship between changing climate and wheat production owing to their involvement in food and nutritional security across the globe [262]. The climate-smart (CS) technological approach is mainly emphasizing the adoption of best and suitable agronomic management practices like timely sowing, efficient nutrient, and water management, contingent crop planning, and protective control measures at delicate growth periods of the crop plant [306]. Conservation agriculture (CA) recently immerged as one of the CS technologies to address several challenges of RWCS. Cultivation of wheat under reduced and zero tillage with retention of surface residue is one of the most accepted CS tools by the farmers of South Asia (SA) to combat various abiotic stress [155]. It was also reported that the adoption of CA is more successful in non-rice aerobic crops like wheat, maize, and pulses [307]. CA is a useful CS technology to combat drought stress because it promotes the early planting of winter wheat under RWCS by utilizing the residual soil moisture after immediate harvesting of monsoon [287]. It is reported that the average yield gains of wheat by adopting CA in SA countries are about 11% [308]. Adoption of different CS technologies such as reduced tillage assured crop establishment methods and efficient nutrient management practices recorded up to 23% higher system productivity and profitability as compared to traditional farming practices in the rice-wheat system of Indo-Gangetic Plains (IGP) of South-Asia [309]. These also help to mitigate various abiotic stresses and reduction of 40% carbon footprint as compared to traditional farmer's practices.

## 5. Summary and Conclusions

Wheat is the most important staple cereal after rice. The demand for wheat is steadily rising day by day in both developed and developing countries due to the food demand of an increasing population. It has been estimated that to encounter the increasing demand for the ever-increasing population of the world, wheat production in the developing world might be increased by 60% in 2050. At the same time, the incidence of abiotic stresses such as drought, salinity, heat, and cold stresses reduce wheat production by 20–30%, particularly in developing countries. Therefore, to alleviate the adverse on wheat productivity, novel approaches are desirable for sustainable wheat production to ensure food and nutritional security of the increasing population. The crop cultivars with better adaptability against abiotic stresses would enhance crop productivity under harsh environments. Therefore, strengthening drought and high temperature tolerance mechanisms is essential for modern varieties, as is the ability to resist against abiotic stresses as well as diseases and pests [310]. The development of stress-tolerant wheat cultivars by mobilizing global biodiversity and using molecular breeding, speed breeding, genetic engineering, and gene editing approaches such as CRISPR-Cas toolkit is desirable for sustainable food production in the modern era of climate change. Many modern varieties have very effective stomatal inhibition of photosynthesis, improved water efficiency, but also better stability of PSII reaction centers and effective energy dissipation mechanisms protecting photosystems

from damage [311–313]. Long-term stress requires a lot of energy to repair individual components and plants are depleted. It turns out that research into the mechanisms of photosynthesis and sucrose metabolism will need to be integrated into breeding technologies. Gradually, modern phenotyping technologies are incorporated into breeding, which offers high accuracy in knowledge not only of growth production parameters but also in flexibility to climatic extremes [314]. Furthermore, the adoption of improved agronomic practices, nanotechnology, and other climate-smart agricultural technologies would help to mitigate climate change-induced abiotic stresses in sustainable wheat production.

**Author Contributions:** Conceptualization of the contents of the review, A.H., M.S., M.B., S.M., S.S. (Sukamal Sarkar) and T.I.; writing—original draft preparation, A.H., S.M., S.S. (Sukamal Sarkar), M.A.A., M.A.S., J.H., S.S. (Saikat Saha), P.B., T.S. and R.B.; writing—review and editing, M.S., M.B., A.K.C., A.E.S. and T.I.; APC funding acquisition, A.H., M.S. and M.B. All authors have read and agreed to the published version of the manuscript.

**Funding:** It is a collaborative work. The APC of the review was supported by Bangladesh Wheat and Maize Research Institute, Bangladesh and the project QK1910343—New Wheat Characters to Improve Adaptation Potential in Global Change Environment and the Slovak University of Agriculture', Nitra, Tr. A. Hlinku 2949 01 Nitra, Slovak Republic under the project 'APVV-18-0465 and EPPN2020-OPVaI-VA-ITMS313011T813'.

**Conflicts of Interest:** The authors declare no conflict of interest. The funders had no role in the design of the study; in the collection, analyses, or interpretation of data; in the writing of the manuscript, or in the decision to publish the results.

## Abbreviations

| | |
|---|---|
| AA | Ascorbic acid |
| ABA | Abscisic acid |
| APX | Ascorbate peroxidase |
| ASH | Ascorbate |
| ASH-GSH | Ascorbate–glutathione |
| BC | Backcross populations |
| CAT | Catalase GR glutathione reductase |
| CIMMYT | International Maize and Wheat Improvement Centre |
| CKs | Cytokinins |
| CRISPR-Cas | Clustered regularly interspaced short palindromic repeats |
| CSSLs | Chromosomal segment substitution lines |
| CWR | Crop Wild Relatives |
| DEPS | Degree of deep oxidation |
| DH | Doubled haploids |
| DHAR | Dehydro ascorbate reductase |
| DNA | Deoxyribonucleic acid |
| ET | Ethylene |
| FAO | Food and Agriculture Organization |
| GA | Gibberellic acid |
| GB | Glycine betaine |
| GR | Glutathione reductase |
| GS | Stomatal conductance |
| GSH | Glutathione |
| GSSG | Oxidized glutathione |
| GST | Glutathione-s-transferase |
| GWAS | Genome-wide association studies |
| IAA | Indole-3-acetic acid |
| IBA | Indole-3-butyric acid |
| ICARDA | International Centre for Agricultural Research in Dry Areas |

| | |
|---|---|
| IPCC | Intergovernmental Panel on Climate Change |
| JA | Jasmonic acid |
| MDA | Malondialdehyde |
| MDHAR | Monodehydro ascorbate reductase |
| MPP | Multiparent populations |
| MTAs | Marker-trait associations |
| MWCNT | Multi-walled-carbon nanotube |
| NAM | Nested association mapping |
| NILs | Near-isogenic lines |
| NPQ | Non-photochemical quenching |
| NPs | Titanium dioxide nanoparticles |
| OECD | The Organisation for economic co-operation and development |
| PAL | Phenylalanine ammonialyase |
| PGPR | Plant growth-promoting rhizobacteria |
| PN | Net assimilation |
| POX | Peroxidase |
| PPO | Polyphenol oxidase |
| QTL | Quantitative trait loci |
| RILs | Recombinant inbred lines |
| RNA | Ribonucleic acid |
| ROS | Reactive oxygen species |
| RWCS | Rice-wheat cropping systems |
| SA | Salicylic acid |
| SOD | Superoxide dismutases |
| VAM | Vesicular-arbuscular mycorrhiza |
| WUE | Water use efficiency |

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
