# Peer review of "Consequences and Mitigation Strategies of Abiotic Stresses in Wheat (Triticum aestivum L.) under the Changing Climate"

_agronomy, doi:10.3390/agronomy11020241_

Round 1
Reviewer 1 Report
In this paper the Authors deeply describes the effect of multiple abiotic stresses on wheat performances.
The review studied all aspects connected to the multiples abiotic stresses on wheat production. The physiological response to drought stress, salinity stress and climate change with particular attention to increased temperature are well deepened. The topic is a complete examination of all aspects related to abiotic stresses and the relative biochemical adaptive mechanisms. The chapters dedicated to the antioxidant production and the phytohormones in response to abiotic stresses are complete and well written. Besides the review analyzed the role of PGPM during abiotic stress with a very rich references. The chapters dedicated to conventional breeding, new breeding technology, germplasm selection and genetic engineering report in detail the last results on wheat breeding. The paper report also the theme of the application of the nano-technologies devices to help in real-time wheat crop growth and information for precision farming in wheat production. The conclusions according me are very robust and all chapter of this review are well presented.Author Response
Response to Reviewer-1 comments
Comments and Suggestions for Authors
In this paper the Authors deeply describes the effect of multiple abiotic stresses on wheat performances.
The review studied all aspects connected to the multiples abiotic stresses on wheat production. The physiological response to drought stress, salinity stress and climate change with particular attention to increased temperature are well deepened. The topic is a complete examination of all aspects related to abiotic stresses and the relative biochemical adaptive mechanisms. The chapters dedicated to the antioxidant production and the phytohormones in response to abiotic stresses are complete and well written. Besides the review analyzed the role of PGPM during abiotic stress with a very rich reference. The chapters dedicated to conventional breeding, new breeding technology, germplasm selection and genetic engineering report in detail the last results on wheat breeding. The paper report also the theme of the application of the nano-technologies devices to help in real-time wheat crop growth and information for precision farming in wheat production. The conclusions according me are very robust and all chapter of this review are well presented.
Authors’ response: Thanks a lot for your valuable comments after reviewing the manuscript and these really provided us enthusiasm to fine-tune it further. Accordingly, some modifications have been made for a greater clarity. Once again, thanks a lot.

Reviewer 2 Report
Literature review is a good tool for reserchers starting in a new field to get a quick overview in the target scientific branch. Therefore it must be clear, understandable, concise, not omitting important but not diving in details. This review was written by several people with different styles, different English, different depth. It would be beneficial for the text when authors would make a revision of the text among themselves.
Abstract: The abstract should address all or at least most of chapters in the main text. It should tell reader what is the article about in a very concise form. The highlighted part part is suitable for introduction not for abstract. The abstract is necessary to rewrite.
The main text: missing chapter 2? Chapter 3.1-3.4 There are often not clear sentences and some are gramatically incorrect. Many sentences are badly constructed, object of the main sentence is used as subject of the secondary sentence and it is not expresed. Finaly it does not make a sense or reader feels lost - confused. e.g. line 220-221, 226-228. Chapter 3.5 is better, however it needs some clarification, reformulation of unclear and long sentences. Beginning pf 3.6 the text is very good and clear, good organized.
All my comments were included in the text.
The submission is publishable after correction, reformulations and clarifications. It can be a useful tool for orientation in the abiotic stress problematic. English correcting of the fist half is needed by native English speaker.

Author Response
Response to Reviewer-2 comments
Comments and Suggestions for Authors
Literature review is a good tool for researchers starting in a new field to get a quick overview in the target scientific branch. Therefore, it must be clear, understandable, concise, not omitting important but not diving in details. This review was written by several people with different styles, different English, different depth. It would be beneficial for the text when authors would make a revision of the text among themselves.
Authors’ response: You are right and there is scope for fine-tuning of the content. Thanks for your valuable suggestion for correction and changes gave been made accordingly. We must communicate our regards for your sincere efforts for improvement of the article.
Abstract: The abstract should address all or at least most of chapters in the main text. It should tell reader what is the article about in a very concise form. The highlighted part is suitable for introduction not for abstract. The abstract is necessary to rewrite.
Authors’ response: Thanks for your good suggestion. The Abstract has been re-arranged.
The main text: missing chapter 2? Chapter 3.1-3.4 There are often not clear sentences and some are grammatically incorrect. Many sentences are badly constructed, object of the main sentence is used as subject of the secondary sentence and it is not expressed. Finally, it does not make a sense or reader feels lost - confused. e.g., line 220-221, 226-228.
Authors’ response: Thanks a lot. As per your comments, necessary changes have been made.
Chapter 3.5 is better; however, it needs some clarification, reformulation of unclear and long sentences.
Beginning pf 3.6 the text is very good and clear, good organized.
Authors’ response: Thanks for your valuable comments and complements. The subheadings are further modified.
Response to comments as per pdf file
The submission is publishable after correction, reformulations and clarifications. It can be a useful tool for orientation in the abiotic stress problematic. English correcting of the first half is needed by native English speaker.
Authors’ response: The article has been modified and limitations identified have been changed/improved to make it appropriate and full-proof. It is, further, for your kind consideration.
Our Replies [The line number mentioned as per the reviewer’s comment; pdf file attached by the reviewer.]
Line 5: delete comma in upper two indexes
Authors’ response: As per suggestion, upper comma deleted.
Lines 44-51: The abstract should address all or at least most of chapters in the main text. It should tell reader what is the article about in a very cocise form.
Authors’ response: We have fully rewritten the Abstract
Line 72: replace calories with Joules
Authors’ response: Energy is often expressed as the calorie (cal). One calorie is equal to 4.184 joules. The Calorie (Cal) is used to express the energy in food instead of jule.
Responses: The line number mentioned as per the reviewer’s comment in pdf file
Authors’ response: Line 90: Changed and as per suggestion it has been made ‘developed’.
Authors’ response: Line 91: As per suggestion it has been made ‘number of grains per plant’.
Authors’ response: Line 94: Modified as per the suggestion and made it as ‘the temperature above ‘.
Authors’ response: Line 95: Modified. The word ‘prevailing’ has been changed into ‘current’ as per the suggestion.
Authors’ response: Line 97: Deleted the word ‘loss’.
Authors’ response: Line 118: It will be sub-heading ‘2’, by mistake it has been done ‘3’. Extremely sorry for the error.
Authors’ response: Line 121: ROS: made plural.
Authors’ response: Line 123: modified as: ‘can lead to….’
Authors’ response: Line 125: Chloroplast has been made plural as ‘chloroplasts’
Authors’ response: Line 131: The word machinery has been replaced by ‘system’; it should be ‘damaged’ [Extremely sorry for the error.]
Authors’ response: Line 133: Modified as ‘causing an oxidative damage’.
Authors’ response: Line 137: the word ‘the’ deleted as per the suggestion.
Authors’ response: Line 138: Modified.
Authors’ response: Line 144: “*” deleted.
Authors’ response: Line 145: Included ‘that’; ‘the’ has been replaced by ‘a’; ‘lead’ has been modified as ‘leading’.
Authors’ response: Line 149: ‘stress induced’ deleted.
Authors’ response: Line 151-152: Modified. ‘Under stressed conditions, pants are acclimatized and changes the usual metabolic pathway with respect to the changes in climates.’
Authors’ response: Line 153: ‘adjustment’ has been replaced by ‘pressure’.
Authors’ response: Line 177: done, as suggested.
Authors’ response: Line 182: Modified as per suggestion.
Authors’ response: Line 188: ‘substrates’ has been replaced by ‘substances’.
Authors’ response: Line 190: The word ‘against’ has been deleted.
Authors’ response: Line 191: Modified as per suggestion.
Authors’ response: Line 211-212: Modified as per the suggestion.
Authors’ response: Line 213: ‘was’ has been included.
Authors’ response: Line 217: The word ‘that’ included.
Authors’ response: Line 220-221: Modified as per the suggestion.
Authors’ response: Line 226-228: Modified.
Authors’ response: Line 274: The words ‘stress-induced’ deleted.
Authors’ response: Line 277: “*” deleted; modified as ‘plastom’.
Authors’ response: Line 292: modified as suggested (phenolic compounds contents). It is ‘closing’.
Authors’ response: Line 299: “GR”, deleted. Extremely sorry!
Authors’ response: Line 300: ‘plants’ has been replaced by ‘plant’.
Authors’ response: Line 306: Corrected. Extremely sorry!
Authors’ response: Line 315: Corrected. Extremely sorry!
Authors’ response: Line 325: Corrected.
Authors’ response: Line 326: Corrected.
Authors’ response: Line 327: Corrected, as suggested.
Authors’ response: Line 329: The word ‘manipulating’ deleted.
Authors’ response: Line 330: Modified as suggested.
Authors’ response: Line 332: Modified as suggested.
Authors’ response: Line 342: Modified as ‘GA’; changed into ‘seed germination’.
Authors’ response: Line 344: The word ‘plant’ has been made ‘plants’; ‘subject to’ deleted; made ‘conditions’.
Authors’ response: Line 351: The word ‘more’ has been replaced by ‘higher’.
Authors’ response: Line 353: The word ‘applied’ has been replaced by ‘application of’.
Authors’ response: Line 354: ‘enhancing percent germination’ has been changed to ‘enhancing germination ability’; the word ‘the’ deleted.
Authors’ response: Line 355: The word ‘crops’ deleted.
Authors’ response: Line 356: ‘also’ has been replaced by ‘as’.
Authors’ response: Line 357: ‘Another, phytohormone cytokinins (CKs)’ replaced by ‘Other phytohormones’.
Authors’ response: Line 358: ‘are’ added.
Authors’ response: Line 362: ‘It also acts a’ modified as ‘It also has a’, as suggested.
Authors’ response: Line 366: Modified. The words ‘and other plastids containing cells’ deleted.
Authors’ response: Line 372: ‘restriction’ modified as ‘controlling’.
Authors’ response: Line 374: ‘tolerance ability’ modified as ‘tolerance to drought stress’.
Authors’ response: Line 376: ‘performs as’ modified as ‘serves as’.
Authors’ response: Line 380: ‘stress ability’ modified as ‘stress tolerance of‘.
Authors’ response: Line 403-404: ‘remain related 404 with the host plant remotely’ modified as ‘rather may occupy host plants on surface’.
Authors’ response: Line 407-413: Modified, as suggested.
Authors’ response: Line 438: ‘hexaploid, breeding’ modified as ‘hexaploidy nature’
Authors’ response: Line 440: ‘wild family members’ modified as ‘crop wild relatives (CWR)’
Authors’ response: Line 450: ‘populace’ replaced as ‘population’
Authors’ response: Line 451: ‘populaces’ replaced as ‘populations’
Authors’ response: Line 452: ‘populaces’ replaced as ‘populations’
Authors’ response: Line 452-453: Modified.
Authors’ response: Line 454: ‘populaces’ replaced as ‘populations’.
Authors’ response: Line 455: ‘blossoming’ has been changed as ‘flowering’.
Authors’ response: Line 456: ‘populace’ replaced as ‘population’.
Authors’ response: Line 457-458: Highlighted portion deleted.
Authors’ response: Line 473: Corrected.
Authors’ response: Line 549: Corrected.
Authors’ response: Line 564: The words ‘genomics studies’ have been modified as ‘genomic studies’ as suggested by the reviewer.
Authors’ response: Line 564-65: Made plural.
Authors’ response: Line 585-592: Sentences have been simplified, as suggested.
Authors’ response: Line 602: Modified. After Line 607, two sentences are included with one reference [299].
[299] Chen, Z., Shen, Z., Zhao, D., Xu, L., Zhang, L., & Zou, Q. Genome-Wide Analysis of LysM-Containing Gene Family in Wheat: Structural and Phylogenetic Analysis during Development and Defense. Genes, 2021, 12(1), 31.
Authors’ response: Line 691: ‘domesticated’ has been replaced by ‘cultivated’; ‘(CWR)’ included.
Authors’ response: Line 722: Made it ‘Silicon (Si)’.
Authors’ response: Line 833: Made it ‘zinc (Zn)’.
Authors’ response: Line 836: Made it ‘Copper (Cu)’
Authors’ response: Line 837: Made it ‘boron (B)’.
Authors’ response: Line 924: Abbreviations alphabetically arranged.
AA Ascorbic acid
ABA Abscisic acid
APX Ascorbate peroxidase
ASH Ascorbate
ASH-GSH Ascorbate–glutathione
BC Backcross populations
CAT Catalase GR Glutathione reductase
CIMMYT International maize and wheat improvement centre
CKs Cytokinins
CRISPR-Cas Clustered regularly interspaced short palindromic repeats
CSSLs Chromosomal segment substitution lines
CWR Crop Wild Relatives
DEPS Degree of deep oxidation
DH Doubled haploids
DHAR Dehydro ascorbate reductase
DNA Deoxyribonucleic acid
ET Ethylene
FAO Food and agricultural organization
GA Gibberellic acid
GB Glycine betaine
GR Glutathione reductase
GS Stomatal conductance
GSH Glutathione
GSSG Oxidized glutathione
GST Glutathione-s-transferase
GWAS Genome-wide association studies
IAA Indole-3-acetic acid
IBA Indole-3-butyric acid
ICARDA International centre for agricultural research in dry areas
IPCC Intergovernmental panel on climate change
JA Jasmonic acid
MDA Malondialdehyde
MDHAR Monodehydro ascorbate reductase
MPP Multiparent populations
MTAs Marker-trait associations
MWCNT Multi-walled-carbon nanotube
NAM Nested association mapping
NILs Near-isogenic lines
NPQ Non-photochemical quenching
NPs Titanium dioxide nanoparticles
OECD The Organisation for economic co-operation and development
PAL Phenylalanine ammonialyase
PGPR Plant growth-promoting rhizobacteria
PN Net assimilation
POX Peroxidase
PPO Polyphenol oxidase
QTL Quantitative trait loci
RILs Recombinant inbred lines
RNA Ribonucleic acid
ROS Reactive oxygen species
RWCS Rice-wheat cropping systems
SA Salicylic acid
SOD Superoxide dismutases
VAM Vesicular-arbuscular mycorrhiza
WUE Water use efficiency

Reviewer 3 Report
I would like to congratulate the Authors for submitting such an interesting review. However, there are certain points that have to be revised or corrected.
First of all, the they give no information at all on the beneficial role of cytogenetics in producing new bread wheat cultivars, resistant to drought. The exploitation of the 1BL.1RS wheat-rye translocation in wheat programs has been proved beneficial, if it is present in good wheat genetic background, in improving not only drought resistance but also the resistance to biotic stress conditions as stem rust, septoria etc. If good allele composition is used there is also no negative effect of the presence of the translocation on quality. Furthermore, they mention hybrids which are not so widely used in bread wheat, mainly due to the existing difficulties in obtaining hybrid seed. If they know any article dealing with bread wheat hybrids and how they are produced, they must include it to avoid misleading information. They must also refer the disadvantages of wheat hybrids if any (do not forget what has happened in India with rice hybrids).
Regarding bread wheat anther culture, it is well established the in this crop that spontaneous chromosome doubling is common and no need of artificial doubling is needed. For this, the sentence in line 681 must be corrected to …by chromosome doubling, spontaneous or artificial. They have also to mention all other approaches which are in use to produce haploid plants in bread wheat especially in recalcitrant bread wheat cultivars.
The last point that must be added is a reference on the existing ethical problems in using biotechnological approaches.
In lines 169 and 171 it is written …observed by [number of references]. It may be better to name the author instead of writing by
In line 177 it is written …lengths, s fresh and dry weights. The letter s Infront of fresh must be deleted.
In the list of references at the end of the article 406 articles are presented. However, in the text the articles from 33 to 132 are missing. This must be faced.
Author Response
Response to Reviewer-3 comments
Comments and Suggestions for Authors
I would like to congratulate the Authors for submitting such an interesting review. However, there are certain points that have to be revised or corrected.
Authors’ response: Thank you very much for your valuable comments after thorough checking of the article. Authors are grateful to you and revised the article as suggested.
First of all, the they give no information at all on the beneficial role of cytogenetics in producing new bread wheat cultivars, resistant to drought. The exploitation of the 1BL.1RS wheat-rye translocation in wheat programs has been proved beneficial, if it is present in good wheat genetic background, in improving not only drought resistance but also the resistance to biotic stress conditions as stem rust, septoria etc. If good allele composition is used there is also no negative effect of the presence of the translocation on quality. Furthermore, they mention hybrids which are not so widely used in bread wheat, mainly due to the existing difficulties in obtaining hybrid seed. If they know any article dealing with bread wheat hybrids and how they are produced, they must include it to avoid misleading information. They must also refer the disadvantages of wheat hybrids if any (do not forget what has happened in India with rice hybrids). Regarding bread wheat anther culture, it is well established the in this crop that spontaneous chromosome doubling is common and no need of artificial doubling is needed. For this, the sentence in line 681 must be corrected to …by chromosome doubling, spontaneous or artificial. They have also to mention all other approaches which are in use to produce haploid plants in bread wheat especially in recalcitrant bread wheat cultivars.
Authors’ response: The above points are addressed in 2.5 and three references are added.
In lines 169 and 171 it is written …observed by [number of references]. It may be better to name the author instead of writing by
Authors’ response: Removed as suggested.
In line 177 it is written …lengths, s fresh and dry weights. The letter s Infront of fresh must be deleted.
Authors’ response: Removed as suggested.
In the list of references at the end of the article 406 articles are presented. However, in the text the articles from 33 to 132 are missing. This must be faced.
Authors’ response: In the present revised article, no omission references are there. [248-250] and [302] are new references cited while reviewed the article.

Round 2
Reviewer 2 Report
The authors corrected the text and accepted suggestions for reformulation. The Abstract is new and better and acceptable.
In the text, there are still some mistakes that need to be corrected (line 159). Formal mistake - oxygen anion 2- must be shown as upper index ( line 299, 333, 343 and possibly elsewhere). Latin taxa Triticeae, Poaceae, Gramineae must be in italics (line 478-481). The paragraph (lines 654-660) is still too complicated to readers, missing subject (who are well equiped?). Anyhow missing subject was more frequent problem of the text but it was already corrected. The English looks good, but still there are expressions that could be verified. I think the text should be checked for English by a native speaker.
Currently the submission can be accepted after minor revision.

Author Response
Reviewer-2 second-round comments
Comments and Suggestions for Authors
The authors corrected the text and accepted suggestions for reformulation. The Abstract is new and better and acceptable.
Authors’ response: Thanks a lot for your valuable comments.
In the text, there are still some mistakes that need to be corrected (line 159).
Authors’ response: in Line 159 corrected as ‘plants’
Formal mistake - oxygen anion 2- must be shown as upper index (line 299, 333, 343 and possibly elsewhere).
Authors’ response: oxygen anion 2, corrected as O2-
Latin taxa Triticeae, Poaceae, Gramineae must be in italics (line 478-481).
Authors’ response: Latin taxa Triticeae, Poaceae, Gramineae have been modified as italic
The paragraph (lines 654-660) is still too complicated to readers, missing subject (who are well equiped?).
Authors’ response: Sentences in lines 654-660 have been refreshed
Anyhow missing subject was more frequent problem of the text but it was already corrected. The English looks good, but still there are expressions that could be verified. I think the text should be checked for English by a native speaker.
Authors’ response: Thanks a lot for your valuable comments. Besides these above corrections, we also thoroughly checked the whole manuscript and have corrected language where necessary.
